# Importance and implementation of safe nursing behaviors in nursing students' clinical practice: Importance–Performance Analysis (IPA), the borich needs assessment model, and the locus for focus model

Eunjung Kim[1], Hae Ran Kim[2]*

1 Department of Nursing, Honam University, Gwangju, Republic of Korea, 2 Department of Nursing, College of Medicine, Chosun University, Gwangju, Republic of Korea

☺ These authors contributed equally to this work.
* rahn00@chosun.ac.kr

## Abstract

### Background

Nursing students are exposed to diverse occupational risks during clinical practicum, which may threaten both their safety and patient safety. Despite the increasing emphasis on safety education following the Patient Safety Act in South Korea, few studies have examined their performance and perceived importance of safe nursing behaviors. This study addresses this gap to provide evidence for improving nursing safety education.

### Methods

A cross-sectional survey was conducted among 160 nursing students from two universities in South Korea who had completed their clinical practicums. Data were collected using a self-administered 29-item questionnaire developed through a literature review and expert validation. It comprises four domains: infection prevention, musculoskeletal injury prevention, chemical hazard prevention, and psychological injury prevention. Data analysis was performed using SPSS/WIN 28.0, including descriptive statistics, paired-sample t-tests, independent t-tests, and one-way ANOVA. Priority analysis was conducted using Importance-Performance Analysis (IPA), the Borich Needs Assessment model, and the Locus for Focus model.

### Results

Importance scores were significantly higher than performance scores across most items, except for "Dispose used ampoules/needles without recapping" and "Dispose sharps into puncture-resistant containers." IPA identified 14 items in the "Keep up the

**Data availability statement:** All relevant data are within the paper and its Supporting Information files.

**Funding:** Ths study was supported by research funding from Chosun University, 2025.

**Competing interests:** The authors declare no competing interests.

good work" quadrant and four items in the "Concentrate here" quadrant. The Borich Needs Assessment model ranked "Know the location of spill kits," "Use devices to reduce musculoskeletal load" and "Use of goggles when biological exposure is possible" as having the highest priorities. The Locus for Focus model classified five items, including personal protective equipment (PPE) use during biological exposure, maintain posture during ergonomic risks, chemical education, and respect for personal dignity, as the top priority (HH quadrant) for educational intervention.

## Conclusions

The assessment tool effectively evaluated nursing students' safe nursing behaviors during clinical practicums. Four domain-specific priorities were identified: personal protective equipment (PPE) use in infection prevention, ergonomic safety practices, chemical safety knowledge, and psychological safety measures. These findings provide evidence-based guidance for designing targeted and practical nursing safety education programs to improve nursing students' preparedness, safety, and patient safety.

## Introduction

Nurses are the largest group of healthcare professionals in hospital medical systems and perform essential medical labor [1], Therefore, nurse safety is closely linked to patient safety. Previous studies have shown a significant correlation between nurses' work environments, burnout, psychological risks, and patient safety. Nurses working in hospital settings are exposed to hazardous environments [2,3]. Needle-stick injuries occur frequently among nurses, with an incidence rate of 42.8% [4]. Additionally, over 90% of nurses experience musculoskeletal symptoms due to prolonged standing (more than four hours) during their work shifts (68.6%) and repetitive hand movements (48.5%) [5]. Nurses also suffer psychological harm due to verbal and physical violence from patients and their guardians [6]. During the COVID-19 pandemic, it was challenging for nurses to safely perform nursing care due to a lack of systematic education on infection management guidelines, supplies, and workforce management. This underscores the importance of addressing these issues by developing nursing experts [7,8]. It is necessary to cultivate nursing professionals equipped with the competencies required for safe nursing management through systematic and consistent education.

Through clinical practice, nursing students enhance their proficiency in essential behaviors, skills, and knowledge required in nursing practice [9]. Clinical practicum experiences can affect nursing students' learning outcomes and future job choices; therefore, the quality and management of clinical practicum environments are critical [10]. However, similar to professional nurses, nursing students face various health risks in clinical settings, including physical injuries from manual labor, biological harm from blood-borne and respiratory infections, chemical exposure, and psychological

damage from emotional labor and patient violence [11–13]. Due to their lack of experience and expertise, nursing students are continuously exposed to harmful clinical environmental factors [5,12]. Understanding safe nursing behaviors to prevent such hazards is crucial, both for maintaining a safe practicum environment for nursing students and for enhancing patient safety.

In South Korea, the Patient Safety Act was enacted in 2015 and implemented in 2017. The Act strengthened the patient safety evaluation criteria and raised awareness of their importance [14]. In addition, the safety management skills of nursing students who interact directly with patients have become significant [15]. Consequently, 'Clinical Practicum Safety Management' was included as a student domain indicator in the nursing education accreditation of nursing colleges, further highlighting the growing importance of safety management. Nursing colleges have established regulations to ensure safe clinical practicum operations for nursing students, specifying safety management measures and emphasizing safety education to prevent accidents [16]. However, most studies on nursing students' safety behaviors in Korea focus on knowledge and confidence in patient safety rather than on the performance or importance of safe nursing behaviors toward patients.

Performance and importance of safe nursing behaviors have been analyzed in several ways. IPA was introduced by Martilla and James [17]. It has been widely used in nursing and also in business, psychology, and education. Compared with other analytical techniques, IPA is easy to use, visually intuitive, and helps prioritize tasks to determine which should be addressed first. IPA quantitatively evaluates the relative importance and performance of a set of attributes to diagnostically observe user experiences [18]. The process has been validated through various nursing-related studies [19,20]. However, it has limited discriminative ability regarding priorities when there are many items or small differences between item levels. To overcome this limitation, the Borich Needs Assessment and Locus for Focus models are used concurrently [21]. The Borich Needs Assessment model ranks priorities by summing the differences between importance and performance and weighting them by importance. This allows the analysis to discriminate between items to identify priorities. The Locus for Focus model is a visual prioritization method that uses a coordinate plane, where the first quadrant, which shows both a high required level and a large gap between levels, indicates the highest priority [22]. Both the Borich Needs Assessment and Locus for Focus models are widely used analytical methods in education and other fields [23].

This study aimed to assess how nursing students perceive the importance of safe nursing behaviors during a clinical practicum and how much they actually perform these behaviors. Furthermore, by utilizing IPA, the Borich Needs Assessment model, and the Locus for Focus model, this study intended to objectively analyze safe nursing behaviors using various methods. In doing so, the study aimed to provide foundational data to inform strategies for nursing safety education.

## Materials and methods

### Design and participants

This study adopted a cross-sectional survey design to analyze the importance and implementation of safe nursing practices among nursing students in clinical practice. Data collection was conducted from January 1, 2024, to May 30, 2024. Before initiating data collection, the study objectives were explained to the responsible authorities at each participating institution and their consent to proceed was obtained.

The participants were nursing students enrolled at two universities in Gwangju Metropolitan City, South Korea. In accordance with the national nursing education accreditation standards in South Korea, all nursing students are required to complete hospital-based clinical practicums during their third and fourth academic years and to receive mandatory safety education prior to clinical practice, including training in infection prevention and patient safety.

Convenience sampling was used to recruit patients with clinical practicum experience in a hospital setting. Only students who understood the study purpose and voluntarily agreed to participate via written informed consent were included. Nursing students without clinical practicum experience were excluded. As a result, participants shared a relatively homogeneous baseline in terms of prior safety education and exposure to clinical practice.

The minimum sample size was calculated using G*Power 3.1.9.7 software for a paired-sample t-test, with a significance level of 0.05, statistical power of 95%, and an effect size of 0.3. The minimum number of participants required was calculated as 147. Accounting for an expected dropout rate of approximately 10%, data were collected from 160 students, and a final analysis was conducted using this sample. Data collection was conducted using an online survey platform configured to require completion of all items prior to submission; consequently, no item-level missing data were observed, and all collected responses were included in the final analysis.

## Measures

**Safe nursing behaviors.** The measurement tool for safe nursing behaviors was developed based on an extensive review of the relevant literature and the Korean Occupational Safety and Health Guide. An expert panel ensured the content validity of the instrument, including one nursing professor, two nursing managers with more than 15 years of professional experience, one professor specializing in infectious diseases, and one professor of neurosurgery. Each prospective item was evaluated for its content validity, and only those with a Content Validity Index (CVI) of 0.8 or higher were retained in the final tool. The questionnaire was originally developed and administered in Korean, as all participants were Korean nursing students. For the purpose of publication, the survey items were translated into English and subsequently reviewed and refined by a native English speaker with academic expertise to ensure linguistic clarity and conceptual equivalence with the original Korean version. The finalized English version of the questionnaire is provided as supplementary material (S1 Appendix) to enhance transparency and reproducibility.

The final instrument comprised 29 items categorized into four distinct domains: 14 items addressing infection prevention in healthcare settings, three items related to the prevention of musculoskeletal injuries, three items focusing on the prevention of injuries caused by chemical substances, and nine items concerning the prevention of psychological harm. The finalized English version of the questionnaire is provided as supplementary material (S1 Appendix) to enhance transparency and reproducibility.

The same set of 29 items was used to assess both the perceived importance and the performance of safe nursing behaviors. For each item, importance and performance were measured separately using identical five-point Likert scales with the same response anchors (1 = "strongly disagree" to 5 = "strongly agree"), allowing direct comparison between the two constructs. Higher scores indicated greater perceived importance or higher levels of performance in safe nursing behaviors. Reliability analyses were conducted separately for importance and performance items for the total scale and each domain, and acceptable to high internal consistency was observed (Supplementary S3 Table).

The instrument demonstrated excellent internal consistency, with a Cronbach's alpha coefficient (α) of 0.90.

## Importance-performance analysis

The Prioritization of safe nursing behaviors was evaluated using the IPA method. The assessment process consisted of the following steps. First, all participants were surveyed regarding both the perceived importance and actual performance level of each safe nursing behavior item. Second, the mean scores for importance and performance were calculated and plotted on the X-axis and Y-axis of the IPA matrix to determine the relative position of each item. Third, the item priorities were derived based on their quadrant locations within the IPA matrix.

Quadrant I, labeled "Keep up the good work" (KU), represents items with both high importance and high performance, indicating well-maintained and high priority areas. Quadrant II, labeled "Concentrate here" (CH), includes items rated as highly important but with low performance; therefore, CH identifies critical areas requiring focused improvement. Quadrant III, labeled "Low priority" (LP), consists of items with both low importance and low performance. These areas can therefore be addressed gradually. Quadrant IV, labeled "Possible overkill" (PO), refers to items with low importance but high performance, suggesting areas where participants' efforts may be excessive and could be optimized.

 

## Borich needs assessment model

The Borich Needs Assessment was conducted using survey data capturing the perceived importance and current performance for each safe nursing behavior item. The needs score for an item is calculated from the summed differences between the importance and performance levels for all respondents, which is multiplied by the item's mean importance score, and then divided by the total number of responses [24]. By weighting the difference scores based on the items' mean importance levels, this method enhances the discriminative power in determining the priority order among items.

## Locus for focus model

The Locus for Focus model was used to provide a visual representation of the analysis using a coordinate plane (Mink et al., 1979) [22]. For each item, the mean importance level and the mean current performance level were used as cut-off values to divide the coordinate plane into four quadrants. The calculated scores for each item are plotted as individual points on this plane.

Quadrant I (High Discrepancy/High Importance: HH) represents items with above-average discrepancy levels and importance ratings, highlighting the highest-priority areas requiring immediate attention. Quadrant II (High Discrepancy/Low Importance: HL) comprises items with importance ratings below the mean, but discrepancy levels above the mean. This indicates secondary priority areas that require increased performance despite lower perceived importance. Quadrant III (Low Discrepancy/Low Importance: LL) includes items with below-average discrepancy and importance levels, representing the lowest-priority areas. Quadrant IV (Low Discrepancy/High Importance: LH) includes items with importance ratings above the mean but discrepancy levels below the mean. This indicates areas where the current performance is relatively high.

## Statistical analysis

The collected data were analyzed using SPSS/WIN 28.0. software (IBM Corp., Armonk, NY, USA). Descriptive statistics, including frequency, percentage, mean, and standard deviation, were used to investigate participants' characteristics. Tests of normality were conducted using the Kolmogorov-Smirnov and Shapiro-Wilk methods. Both yielded p-values greater than 0.05, indicating that the assumption of normality was satisfied.

The importance and performance levels of safe nursing behaviors, specifically in infection prevention in healthcare, prevention of musculoskeletal injuries, prevention of injuries caused by chemical substances, and prevention of psychological harm, were analyzed using means and standard deviations. Paired sample t-tests were used to assess the differences between importance and performance. Furthermore, differences in importance and performance according to general characteristics were examined using independent t-tests and one-way analysis of variance (ANOVA). In addition, a sensitivity analysis using the Holm–Bonferroni correction was conducted to assess the robustness of the item-level paired comparisons, and the results are presented in Supplementary S2 Table.

For an in-depth analysis of nursing students' safe nursing behaviors regarding importance and performance, IPA, the Borich Needs Assessment model, and the Locus for Focus model were utilized according to the following procedure:

1. Paired t-tests were conducted to determine the differences between current performance and perceived importance levels for each safe nursing behavior item.

2. Priority rankings were produced based on the Borich Needs Assessment model.

3. The results were visually plotted on a coordinate plane following the Locus for Focus model.

4. Items located in the High Discrepancy/High Importance (HH) quadrant of the Locus for Focus model were identified and priority rankings were established accordingly.

5. The top priority group was established by identifying overlapping items between the highest-ranked items from the Borich Needs Assessment model and the HH quadrant of the Locus for Focus model, thus determining the first and second priority groups for targeted intervention.

This combined approach identified a robust, multi-perspective prioritization of safe nursing behaviors, facilitating evidence-based decision-making for nursing safety education strategies.

### Ethical approval

This study was approved by the Institutional Review Board (IRB) of Honam University (Approval No. 1041223–202311-HR-37). Participants were provided with an informed consent form in the first section of the online survey. The consent form explained the study purpose and assured the participants that all collected data would remain confidential and would not be used for any purpose other than research. Participation was strictly voluntary, and the form clearly stated that withdrawal from the study at any time would incur no penalties or disadvantages. These procedures ensured that participants were fully informed of their rights and were able to make autonomous decisions regarding their participation. The consent form was completed before participating in the study. Anonymity was maintained throughout the study, further protecting the confidentiality and privacy of participant information.

## Results

### Participant characteristics

A total of 160 nursing students participated in the study, with a mean age of 23.38 years; participants aged 23 years accounted for 47.5% of the sample. Most were female (83.8%) and in their fourth year of study (75.0%). Regarding religion, 75.6% reported no religious affiliation. Participants reported relatively high self-perceived performance levels of safe nursing behaviors based on global self-assessments of their overall competence within each safety domain. The mean self-perceived performance levels (mean±SD), rated on a five-point Likert scale, were 4.75±0.60 for medical infection prevention, 4.46±0.64 for musculoskeletal injury prevention, 4.47±0.64 for chemical hazard prevention, and 4.53±0.63 for psychological damage prevention (Table 1). These scores reflect participants' subjective, domain-level evaluations of their overall performance and should be interpreted as global self-perceived performance rather than item-level behavioral performance.

Regarding prior educational experiences, 88.1% of the participants reported having previously received education related to safe nursing behaviors; however, 78.8% indicated that they felt the need for additional training. The preferred methods of education were online lectures (61.9%), simulation-based training (16.9%), and practical hands-on training (13.1%), whereas traditional lecture-based education was the least favored (4.4%). In terms of the desired duration, the most preferred format was three one-hour sessions, reported by 71.3% of the participants (Table 1).

### Nursing students' perceptions of the importance and performance of safe nursing behaviors

Paired-sample t-tests were conducted to examine the differences between the perceived importance and actual performance of safe nursing behaviors among nursing students. As shown in Table 2, the importance scores were higher than the performance scores across all domains. However, the differences for the items "Dispose used ampoules/needles without recapping" and "Dispose sharps into puncture-resistant containers" were not statistically significant.

In terms of importance, the highest-rated items (M = 3.98) were "Use disposable gloves for cleaning environment/equipment," "Dispose used ampoules/needles without recapping," "Dispose sharps into puncture-resistant containers," "Wear a mask when having respiratory symptoms," "Receive helpful feedback on practice," and "No discrimination by age, gender, or school." For performance, the highest mean scores were observed for "Dispose used ampoules/needles without

**Table 1. Characteristics of the Participants (N = 160).**

| Characteristics | Categories | n (%) | M±SD |
|---|---|---|---|
| Age (years) | ≤ 22 | 41 (25.7) | 23.38±2.70 |
| | 23 | 76 (47.5) | |
| | ≥ 24 | 43 (26.8) | |
| Gender | Male | 26 (16.3) | |
| | Female | 134 (83.8) | |
| Academic year | 3rd | 40 (25.0) | |
| | 4th | 120 (75.0) | |
| Religion | Yes | 39 (24.4) | |
| | No | 121 (75.6) | |
| Self-perceived performance level | Level of medical infection | | 4.75±0.60 |
| | Level of musculoskeletal prevention | | 4.46±0.64 |
| | Level of chemical prevention | | 4.47±0.64 |
| | Level of psychological damage | | 4.53±0.63 |
| Safe nursing education experience | Yes | 141 (88.1) | |
| | No | 19 (11.9) | |
| Safe nursing education requirements | Yes | 126 (78.8) | |
| | No | 34 (21.3) | |
| Preferred learning method | Lecture | 7 (4.4) | |
| | Online | 99 (61.9) | |
| | Practice | 21 (13.1) | |
| | Simulation | 27 (16.9) | |
| | Virtual reality | 6 (3.8) | |
| Preferred learning time | One hour a day for three days | 114 (71.3) | |
| | Two hours a day for six days | 21 (13.1) | |
| | Two hours a day for a week | 25 (15.6) | |

M = Mean; SD = Standard deviation.

recapping" and "Dispose sharps into puncture-resistant containers" (M = 3.96), followed by "Wear a mask when having respiratory symptoms" (M = 3.93), and "No discrimination by age, gender, or school" along with "Appropriate response to verbal/physical violence" (M = 3.81).

## Importance–performance analysis

The IPA matrix was constructed using the mean importance score (3.93) and performance score (3.70) (Fig 1) According to the IPA results:

- Quadrant 1 (KU) includes items that are ranked high in both importance and performance. These comprised items 2, 3, 5, 9–13, and 23–28, which mainly belong to the domains of infection prevention and psychological injury prevention. This indicates that these areas were well-maintained by nursing students.

- Quadrant 2 (CH) consists of items that are considered highly important but have lower performance scores. The items in this quadrant were: "Use of gown when biological exposure is possible" (Item 7), "Maintain posture during ergonomic risks (e.g., transfers)" (Item 15), "Education on chemical safety" (Item 17), and "Respect for personal dignity" (Item 21). These are critical areas requiring improvement.

**Table 2. Importance and Performance Gaps in Safe Nursing Behaviors (N = 160).**

| Variables | Importance | Performance | Gap | t | p | Borich Needs Assessment Model | | Locus for Focus Model |
|---|---|---|---|---|---|---|---|---|
| | M ± SD | M ± SD | M ± SD | | | Needs | Rank | |
| **I. Infection Prevention** | | | | | | | | |
| 1. Education and information on standard precautions | 3.87 ± 0.34 | 3.66 ± 0.50 | 0.21 ± 0.52 | 5.06 | < 0.001 | 0.81 | 14 | |
| 2. Hand hygiene before and after glove use | 3.96 ± 0.21 | 3.84 ± 0.42 | 0.12 ± 0.43 | 4.38 | < 0.001 | 0.48 | 24 | |
| 3. Hand hygiene after each patient contact | 3.95 ± 0.22 | 3.76 ± 0.47 | 0.19 ± 0.46 | 5.38 | < 0.001 | 0.75 | 16 | |
| 4. Handwashing before leaving the patient room | 3.91 ± 0.29 | 3.70 ± 0.51 | 0.21 ± 0.52 | 5.06 | < 0.001 | 0.82 | 13 | |
| 5. Use of gloves when biological exposure is possible | 3.95 ± 0.22 | 3.81 ± 0.47 | 0.14 ± 0.43 | 4.07 | < 0.001 | 0.55 | 21 | |
| 6. Use of goggles when biological exposure is possible | 3.88 ± 0.33 | 3.41 ± 0.89 | 0.46 ± 0.85 | 6.91 | < 0.001 | 1.78 | 3 | |
| 7. Use of gown when biological exposure is possible | 3.93 ± 0.26 | 3.60 ± 0.65 | 0.33 ± 0.58 | 7.11 | < 0.001 | 1.30 | 5 | HH |
| 8. Avoid contamination when handling soiled linen | 3.93 ± 0.26 | 3.74 ± 0.60 | 0.18 ± 0.56 | 4.10 | < 0.001 | 0.70 | 18 | |
| 9. Use disposable gloves for cleaning environment/equipment | 3.91 ± 0.29 | 3.79 ± 0.44 | 0.11 ± 0.34 | 4.23 | < 0.001 | 0.44 | 25 | |
| 10. Dispose used ampoules/needles without recapping | 3.98 ± 0.14 | 3.96 ± 0.19 | 0.02 ± 0.18 | 1.35 | 0.181 | 0.08 | 27 | |
| 11. Dispose sharps into puncture-resistant containers | 3.98 ± 0.14 | 3.96 ± 0.19 | 0.02 ± 0.18 | 1.35 | 0.181 | 0.08 | 28 | |
| 12. Cover nose/mouth when coughing; hand hygiene after tissue disposal | 3.94 ± 0.24 | 3.81 ± 0.58 | 0.13 ± 0.55 | 2.89 | 0.004 | 0.51 | 23 | |
| 13. Wear a mask when having respiratory symptoms | 3.98 ± 0.14 | 3.93 ± 0.25 | 0.05 ± 0.22 | 2.89 | 0.004 | 0.20 | 26 | |
| **II. Musculoskeletal Injury Prevention** | | | | | | | | |
| 14. Education on musculoskeletal safety | 3.83 ± 0.42 | 3.59 ± 0.68 | 0.24 ± 0.57 | 5.31 | < 0.001 | 0.92 | 10 | |
| 15. Maintain posture during ergonomic risks (e.g., transfers) | 3.94 ± 0.23 | 3.66 ± 0.55 | 0.29 ± 0.52 | 7.01 | < 0.001 | 1.14 | 7 | HH |
| 16. Use devices to reduce musculoskeletal load | 3.80 ± 0.50 | 3.33 ± 0.87 | 0.48 ± 0.78 | 7.74 | < 0.001 | 1.82 | 2 | |
| **III. Chemical Hazard Prevention** | | | | | | | | |
| 17. Education on chemical safety | 3.94 ± 0.29 | 3.64 ± 0.57 | 0.26 ± 0.52 | 6.39 | < 0.001 | 1.02 | 8 | HH |
| 18. Use PPE (apron, rubber gloves) for chemicals | 3.85 ± 0.36 | 3.50 ± 0.83 | 0.35 ± 0.76 | 5.81 | < 0.001 | 1.35 | 4 | |
| 19. Know the location of spill kits | 3.88 ± 0.33 | 3.28 ± 0.90 | 0.60 ± 0.84 | 9.03 | < 0.001 | 2.33 | 1 | |
| **IV. Psychological Injury Prevention** | | | | | | | | |
| 20. Education on psychological safety | 3.88 ± 0.43 | 3.63 ± 0.64 | 0.25 ± 0.66 | 4.84 | < 0.001 | 0.97 | 9 | |
| 21. Respect for personal dignity | 3.94 ± 0.23 | 3.62 ± 0.60 | 0.32 ± 0.59 | 6.87 | < 0.001 | 1.26 | 6 | HH |
| 22. Positive engagement with peers | 3.91 ± 0.33 | 3.76 ± 0.46 | 0.15 ± 0.47 | 4.08 | < 0.001 | 0.59 | 20 | |
| 23. Receive helpful feedback on practice | 3.98 ± 0.16 | 3.79 ± 0.45 | 0.19 ± 0.44 | 5.43 | < 0.001 | 0.76 | 15 | |
| 24. No discrimination by age, gender, or school | 3.98 ± 0.14 | 3.81 ± 0.48 | 0.17 ± 0.47 | 4.59 | < 0.001 | 0.68 | 19 | |
| 25. Appropriate response to verbal/physical violence | 3.96 ± 0.21 | 3.81 ± 0.48 | 0.14 ± 0.51 | 3.55 | 0.001 | 0.55 | 22 | |
| 26. Appropriate response to unwanted sexual attention | 3.96 ± 0.21 | 3.73 ± 0.66 | 0.23 ± 0.66 | 4.46 | < 0.001 | 0.91 | 11 | |
| 27. Appropriate response to threats | 3.95 ± 0.22 | 3.76 ± 0.62 | 0.19 ± 0.57 | 4.13 | < 0.001 | 0.75 | 17 | |
| 28. Appropriate response to humiliating behavior | 3.97 ± 0.16 | 3.76 ± 0.63 | 0.22 ± 0.63 | 4.38 | < 0.001 | 0.87 | 12 | |

M = Mean; SD = Standard deviation.

- Quadrant 3 (LP) contains items rated low in terms of both importance and performance. These comprised Items 1, 6, 14, 16, and 18–20. This indicates that these behaviors were perceived as less important and were also infrequently performed, suggesting a relatively low priority for immediate intervention.

- Quadrant 4 (PO) includes items with lower importance but relatively higher performance. These were: "Handwashing before leaving the patient room" (Item 4), "Avoid contamination when handling soiled linen" (Item 8), and "Positive

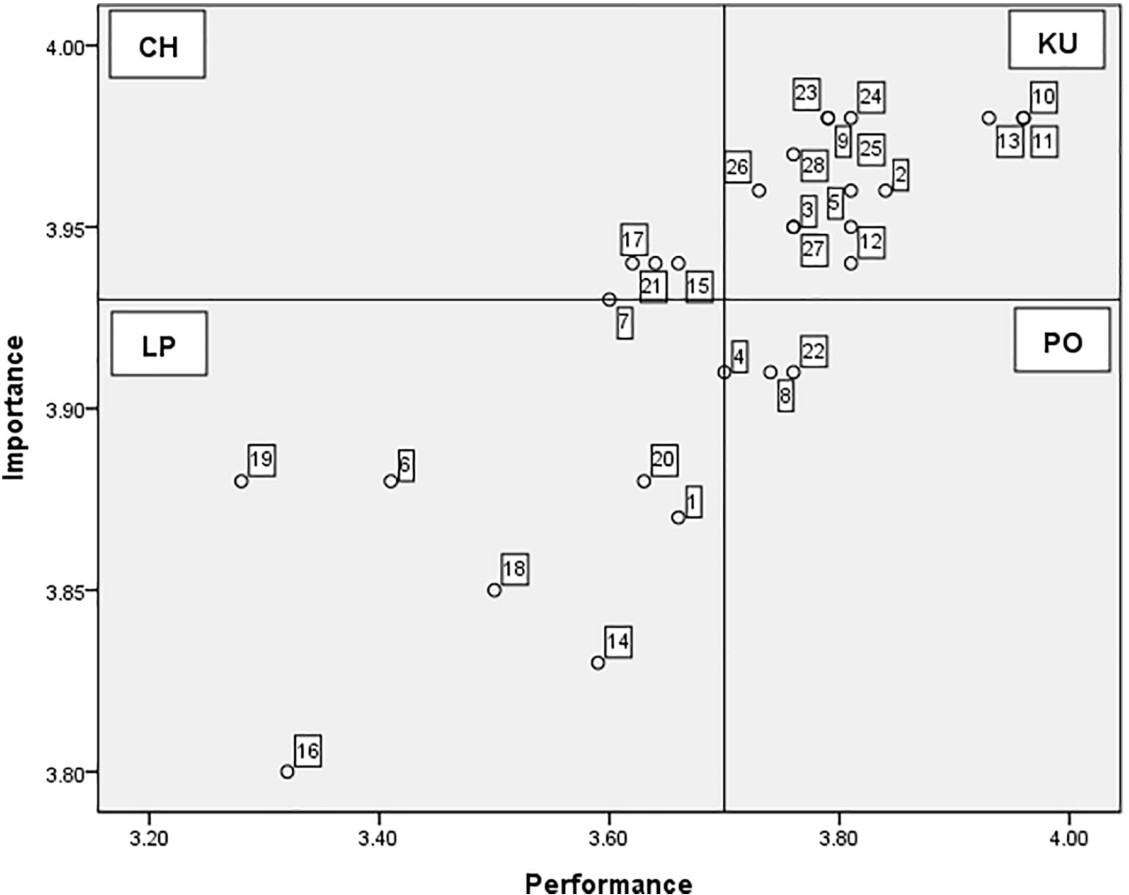

**Fig 1. Importance–Performance Analysis (IPA) of safe nursing behaviors among nursing students.** KU = Keep Up the Good Work (high importance, high performance); CH = Concentrate Here (high importance, low performance); LP = Low Priority (low importance, low performance); PO = Possible Overkill (low importance, high performance).

engagement with peers" (Item 22). This pattern suggests that these behaviors may be performed more than warranted based on their perceived importance, indicating a potential over-allocation of effort or resources.

This IPA matrix provides clear guidance for prioritizing efforts to enhance safe nursing behaviors among nursing students, emphasizing which performance strengths should be maintained and which critical gaps should be addressed (Fig 1).

### Borich needs assessment model

Borich Needs Assessment was used to identify nursing students' educational needs regarding safe nursing behaviors. The model evaluates the discrepancies between perceived importance and actual performance, thereby deriving priority rankings. According to the Borich Needs scores (Table 2), the highest educational priority was "Know the location of spill kits" (Item 19) with a score of 2.33 points. This was followed "Use devices to reduce musculoskeletal load" (Item 16; 1.82 points), "Use of goggles when biological exposure is possible" (Item 6; 1.78 points), and "Use PPE (apron, rubber gloves) for chemicals" (Item 18; 1.35 points) and "Use of gown when biological exposure is possible" (Item 7; 1.30 points).

These results highlight key areas in which nursing students urgently require targeted education to improve their safe nursing behaviors. In particular, education should focus on the awareness and use of protective measures and equipment

to mitigate biological and musculoskeletal risks. This prioritization can inform the development of effective nursing safety curricula and training interventions tailored to students' needs.

**Locus for focus model**

The Locus for Focus model visualization of nursing students' educational priorities regarding safe nursing behaviors is presented in Fig 2. The mean score for educational importance was 3.92 points. The average discrepancy between importance and performance was 0.24 points, which was used as the reference axis for the quadrant divisions.

In the Locus for Focus model, the items located in the first quadrant (High Discrepancy/High Importance, HH) represent priorities requiring the most urgent and focused educational attention. These top-priority items included five safety nursing behaviors: "Use of gown when biological exposure is possible" (Item 7), "Maintain posture during ergonomic risks (e.g., transfers)" (Item 15), "Education on chemical safety" (Item 17), and "Respect for personal dignity" (Item 21). The fourth quadrant (Low Discrepancy/High Importance, LH) comprises items with high importance but relatively small discrepancies in performance, indicating areas that are already well addressed but still important. This comprised Items 2, 3, 5, 9–13, 23–28. The second quadrant (High Discrepancy/Low Importance, HL) included items with lower importance but a notable gap between importance and performance, specifically, Items 6, 14, 16, 18–20. Finally, the third quadrant (Low

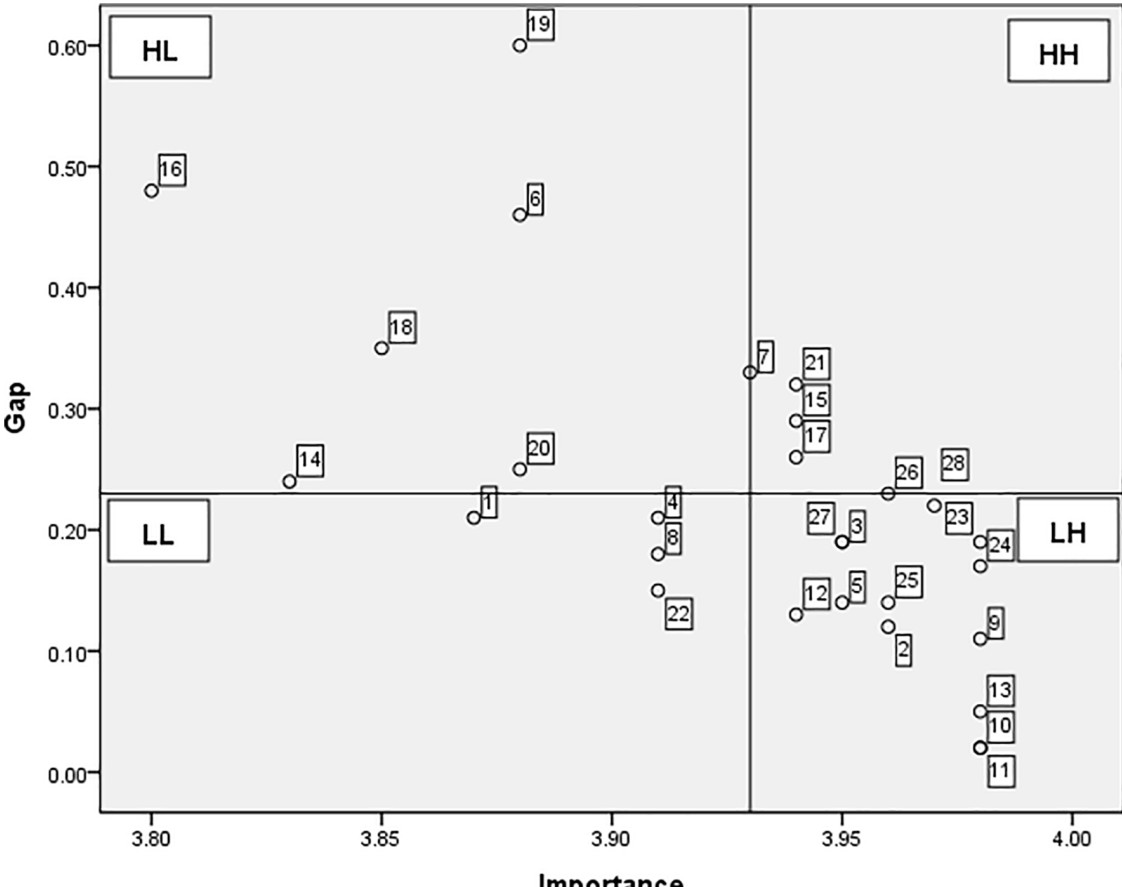

**Fig 2. Locus for Focus Model Analysis of nursing safety education requirements.** HH = High Discrepancy/High Importance; HL = High Discrepancy/Low Importance; LL = Low Discrepancy/Low Importance; LH = Low Discrepancy/High Importance.

Discrepancy/Low Importance, LL) encompassed items with both low importance and low discrepancy, including Items 1, 4, 8, and 22. This Locus for Focus model visualization effectively highlights the critical areas to prioritize nursing safety education. The HH quadrant should be emphasized for immediate intervention and the other quadrants should inform ongoing training strategies (Fig 2).

## Discussion

This study investigated nursing students' safe nursing behaviors during clinical practicums by assessing their performance and perceived importance and identifying areas with significant gaps that require prioritization, areas to maintain, and areas showing potential over performance. The findings clarify the discrepancies between perceived importance and the actual performance of safe nursing behaviors, providing foundational data for developing targeted educational programs to enhance safe nursing practice.

Using IPA, 14 items were classified as belonging to the KU quadrant, indicating both high importance and performance levels (Items 2, 3, 5, 9–13, and 23–28). Four items (7, 15, 17, 21) fell into the CH quadrant, representing behaviors that are highly important but performed inadequately.

Borich Needs Analysis further refined the priorities within these 14 items identified through IPA. Using the Locus for Focus model, five items (7, 13, 15, 17, and 21) were positioned in the highest-priority quadrant (High Discrepancy/High Importance, HH), while six items (6, 14, 16, and 18–20) were in the second-priority quadrant (High Discrepancy/Low Importance, HL).

In the infection prevention domain, top priority items included "Use of gown when biological exposure is possible" and "Use of goggles when biological exposure is possible" ranked as a secondary priority. These results align with existing research indicating low confidence and knowledge among nurses regarding PPE use, as well as low mask-wearing performance among nursing students [25]. Despite heightened awareness of infection control due to infectious disease outbreaks, challenges persist in proper PPE usage. This underscores the need for practical and accurate PPE education starting at the nursing student stage.

In the musculoskeletal injury prevention domain, "maintaining posture during ergonomic risks (e.g., patient transfers)" was identified as the highest priority, followed by "education on musculoskeletal safety" and "use of devices to reduce musculo-skeletal load." In clinical settings, nurses are frequently required to lift heavy objects or reposition patients and perform tasks that place considerable strain on the musculoskeletal system, increasing their vulnerability to pain and injury [26]. Nursing students, in particular, are at elevated risk of these outcomes, as their unfamiliarity with the clinical environment often results in awkward and uncomfortable postures during nursing activities. This may explain nurses' high prioritization of proper posture maintenance. However, educational and informational support regarding musculoskeletal safety in clinical training institutions remains insufficient. Furthermore, the use of assistive devices to reduce musculoskeletal load is relatively uncom-mon. If such conditions persist after nursing students transition to become novice nurses, they may struggle to fully adapt to their roles, experience musculoskeletal pain, develop job-related stress, and ultimately experience an increased likelihood of turnover [27]. Therefore, it is essential to provide continuous education and training on proper body mechanics and posture from the student nurse stage onward. Through regular practice and simulation, nursing students should be encouraged to incorporate preventive techniques into their routine nursing tasks to reduce the risk of musculoskeletal injury.

In the chemical hazard prevention domain, "Education on chemical safety" was ranked as the highest priority, followed by "Use PPE (apron, rubber gloves) for chemicals" and "Know the location of spill kits." This finding is consistent with previous studies reporting that nurses lack adequate training on hazards and safe handling practices. It also aligns with earlier research showing that hospital nurses' preparedness for chemical accident responses was low both before and after education [28,29]. In the clinical setting, hospitals handle a variety of medications, including antineoplastic agents, as well as chemicals for disinfection and cleaning. Nurses frequently prepare hazardous drugs; however, their compliance with PPE guidelines is inconsistent. Moreover, nurses often do not perceive the actual risk of exposure and exhibit low adherence to PPE use [29]. For nursing students, safety education related to chemical substances—such as information

on hazardous chemicals, antidotes, decontamination principles, and procedures—is brief. Furthermore, prior studies in this area are scarce. Therefore, it is necessary to provide nursing students with preemptive information on commonly used hospital chemicals and to implement training that supports nursing students' safety in clinical environments. It should be noted that the performance of certain safety behaviors, such as knowing the location of spill kits and using assistive devices to reduce musculoskeletal load, may be influenced not only by educational factors but also by environmental constraints, including resource availability and institutional policies within clinical settings. In nursing students' clinical practicums, access to and use of such resources are largely determined by affiliated hospitals rather than by students themselves, which may limit performance despite high perceived importance. Therefore, educational gaps and environmental constraints are likely interrelated, and future studies should incorporate contextual variables related to institutional infrastructure and policies to better disentangle these influences.

In the psychological injury prevention domain, "Respect for personal dignity" ranked highest, followed by "Education on psychological safety." This indicates that, while respecting personal dignity is recognized as important, it is not adequately practiced in reality. Clinical practicums in hospitals are an essential component of nursing education in general and for Korean nursing students. However, nursing students experience clinical practicum stress due to unfamiliar practice settings, insufficient proficiency in core nursing skills, lack of experience, complex interpersonal relationships with healthcare staff, patients, and caregivers, and nurses' indifference and non-educational attitudes [30]. Verbal abuse during clinical practicums is prevalent among Korean nursing students (reported at 94.8%), which stems from Korea's hierarchical senior-junior culture and the authoritative, rank-oriented nursing organizational culture [31]. Repeated exposure to such experiences increases clinical practicum stress and depression, while diminishing nursing students' professional self-concepts [32]. Furthermore, in Korea, the shortage of nurses and the growing societal interest in healthcare have led to continuous expansion of nursing school enrollments [33]. Consequently, multiple universities compete for student placements in limited hospital training sites, and nursing students, concerned about securing stable practicum opportunities, may refrain from expressing dissatisfaction even when they recognize unfair treatment [34]. These circumstances negatively impact nursing students' practicum satisfaction and professional identities.

Therefore, it is essential to create a clinical environment that respects nursing students' rights as learners and provides practicum experiences that foster their interest. Moreover, nursing educational institutions such as the Korean Accreditation Board of Nursing Education, hospitals, and university nursing program councils should establish cooperative frameworks to develop appropriate measures for safe nursing behaviors, accompanied by necessary institutional reforms.

### Limitations and strengths

This study had several limitations. First, participants were limited to nursing students from two universities in South Korea. Which may restrict the generalizability of the findings. Therefore, the results should be interpreted with caution when applied to nursing students who have completed clinical practicums in a wider range of institutional and clinical settings, including diverse hospital environments and nursing education contexts. Future research should include nursing students from multiple regions and a broader range of hospitals to enhance the external validity of these findings.

Second, study data were collected using a self-administered questionnaire, which may be subject to response bias and social desirability bias and may not fully capture the complexity and contextual nuances of nursing students' experiences and perceptions. In addition, several performance-related items—such as knowledge of spill kit locations and use of assistive devices to reduce musculoskeletal load—may have been influenced by institutional resource availability and organizational policies, factors that were not directly measured in this study. As a result, it was not possible to fully disentangle educational gaps from environmental or structural constraints. Future studies employing mixed-methods approaches, including qualitative interviews, direct observation, or assessment of institutional safety resources and policies, are recommended to provide a more comprehensive and context-sensitive understanding of safe nursing behaviors among nursing students.

Third, although a large proportion of participants (88.1%) reported having received prior safety education, the substantial imbalance between students with and without such education limited the feasibility of conducting robust subgroup analyses. Consequently, the potential association between prior safety education and reduced importance–performance gaps could not be empirically examined in this study. Future studies with more balanced group sizes or stratified sampling designs are needed to clarify the impact of prior safety education on safe nursing behaviors.

Fourth, although the overall scale demonstrated high internal consistency, one domain (musculoskeletal injury prevention) showed relatively low Cronbach's alpha values, particularly for the importance scale. This may be attributable to the small number of items and the conceptual heterogeneity of behaviors within that domain, as Cronbach's alpha is sensitive to item count. Given that the primary aim of this study was needs assessment and priority setting rather than psychometric scale development, the domain was retained; however, future research should consider expanding item numbers and conducting further psychometric validation.

Fifth, exploratory or confirmatory factor analyses (EFA/CFA) were not conducted to empirically validate the four-domain structure, as the instrument was theoretically defined a priori and content validity was established through expert review. Future studies with larger and more diverse samples are warranted to examine the latent structure of the instrument using factor analytic approaches.

Finally, this study employed a cross-sectional design, which limits causal inference regarding relationships between perceived importance, performance, and identified priority gaps. Longitudinal or intervention-based studies are needed to examine changes in safe nursing behaviors over time and to evaluate the effectiveness of targeted educational programs.

Despite these limitations, this study has notable strengths. To the best of our knowledge, this is the first study in South Korea to develop and apply a comprehensive measurement tool specifically designed to assess safe nursing behaviors among nursing students during clinical practicums, rather than focusing solely on licensed nurses or isolated safety issues. By targeting nursing students as future healthcare professionals, this study addresses a critical yet underexplored stage of professional development in nursing safety research.

A major strength of this study lies in its multidimensional conceptualization of safe nursing behaviors. By integrating infection prevention, musculoskeletal injury prevention, chemical hazard prevention, and psychological injury prevention into a single analytical framework, this study provides a holistic and nuanced understanding of safety behaviors in clinical practice. This approach reflects the complex and multifaceted nature of occupational risks faced by nursing students and extends prior research that has typically examined these domains in isolation.

Another important strength is the application of multiple complementary priority-setting methodologies, including Importance–Performance Analysis (IPA), the Borich Needs Assessment model, and the Locus for Focus model. The triangulation of these analytic approaches enhances the robustness and interpretability of the findings by allowing consistent identification of priority areas across different methodological perspectives. This integrative analytic strategy moves beyond simple mean comparisons and offers practical, evidence-based guidance for educational decision-making.

In addition, the measurement tool was developed through a rigorous process that included an extensive literature review, reference to national occupational safety guidelines, and expert content validation by a multidisciplinary panel. The instrument demonstrated high overall internal consistency, supporting its suitability for needs assessment and applied research in nursing education contexts. Finally, the study provides empirically grounded and actionable evidence that can inform the design of targeted, domain-specific safety education and training programs, with potential implications for improving both nursing students' safety and patient safety in clinical settings.

## Conclusions

This study developed a safe nursing behavior assessment tool for nursing students with clinical practicum experience, examining their performance and the perceived importance of these behaviors. Using a combination of IPA, the Borich

Needs Assessment model, and the Locus for Focus model, the study identified both top- and secondary-priority items for educational intervention.

In the infection prevention domain, wearing protective goggles, gowns, and masks was ranked the highest. In the musculoskeletal injury prevention domain, all items related to musculoskeletal safety, including education, maintaining proper posture, and use of assistive devices, were identified as high priorities. The chemical hazard prevention domain highlighted the need for education on chemical safety, the use of personal protective equipment, and knowledge of spill kit locations. In the psychological injury prevention domain, education on psychological safety and respect for personal dignity were emphasized as key priorities.

These findings demonstrate that the developed tool can be effectively used to assess and evaluate the safety of nursing students in clinical practice. Furthermore, the identification of top and secondary priorities using multiple analytical methods provides evidence-based guidance for the development of targeted training programs. Such programs can be tailored to address specific safety needs and, ultimately, improve nursing students' preparedness to perform safe nursing practices during clinical education.

## Supporting information

**S1 Table. Cronbach_Reliability.**
(DOCX)

**S2 Table. Paired_Comparison_HolmBonferroni.**
(DOCX)

**S3 Table. Prioritization_Borich_IPA_LFF.**
(DOCX)

**S1 Appendix. Questionnaire on Safe Nursing Behaviors.**
(DOCX)

## Author contributions

**Conceptualization:** Eunjung Kim, Hae Ran Kim.

**Data curation:** Eunjung Kim, Hae Ran Kim.

**Funding acquisition:** Hae Ran Kim.

**Investigation:** Eunjung Kim, Hae Ran Kim.

**Methodology:** Eunjung Kim, Hae Ran Kim.

**Supervision:** Hae Ran Kim.

**Writing – original draft:** Eunjung Kim, Hae Ran Kim.

**Writing – review & editing:** Eunjung Kim, Hae Ran Kim.

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
