## [Decision Letter · Decision Letter 0]

9 Dec 2025

Dear Dr. Hae Ran Kim,

Thank you for submitting your manuscript to PLOS ONE. After careful consideration, we feel that it has merit but does not fully meet PLOS ONE’s publication criteria as it currently stands. Therefore, we invite you to submit a revised version of the manuscript that addresses the points raised during the review process.

**ACADEMIC EDITOR:**  Authors are required to reply all the queries, raised by the reviewers.

We look forward to receiving your revised manuscript.

Kind regards,

Priti Chaudhary, M.S.

Academic Editor

PLOS One

Journal Requirements:

https://journals.plos.org/plosone/s/file?id=ba62/PLOSOne_formatting_sample_title_authors_affiliations.pdf ..

5. Please be informed that funding information should not appear in the Acknowledgments section or other areas of your manuscript. We will only publish funding information present in the Funding Statement section of the online submission form. Please remove any funding-related text from the manuscript.

Reviewers' comments:

Reviewer's Responses to Questions

**Comments to the Author**

1. Is the manuscript technically sound, and do the data support the conclusions?

Reviewer #1: Partly

Reviewer #2: Yes

Reviewer #3: Yes

2. Has the statistical analysis been performed appropriately and rigorously?

Reviewer #1: I Don't Know

Reviewer #2: No

Reviewer #3: Yes

3. Have the authors made all data underlying the findings in their manuscript fully available?

Reviewer #1: Yes

Reviewer #2: Yes

Reviewer #3: Yes

4. Is the manuscript presented in an intelligible fashion and written in standard English?

Reviewer #1: Yes

Reviewer #2: Yes

Reviewer #3: Yes

Reviewer #1: Reviewer comment and suggestions

Strengths:

1. Relevance and Significance: The study addresses an important gap in nursing education—assessing safety behaviors of nursing students during clinical practice, which is crucial for both student and patient safety.

2. Methodological Rigor: The use of a validated questionnaire and multiple analytical models (IPA, Borich, Locus for Focus) enhance the robustness of the findings.

3. Comprehensive Framework: The study covers multiple domains (infection, musculoskeletal injuries, chemicals, psychological safety), providing a holistic view of safety behaviors.

4. Practical Implications: The results offer clear, actionable priorities for targeted education programs, potentially improving safety outcomes.

Weaknesses and Limitations:

1. Sample and Generalizability: The sample is limited to 160 students from two universities in South Korea, which may limit broader applicability.

2. Cross-Sectional Design: The design captures perceptions and behaviors at a single point, limiting insights into causality or changes over time.

3. Self-Reported Data: Reliance on self-administered questionnaires may introduce response bias, with students potentially overestimating their safety behaviors.

4. Lack of Qualitative Insights: The study might benefit from qualitative data to explore underlying reasons for gaps in safety behaviors or perceived importance.

5. Limited Contextual Detail: Details such as students' prior safety training, clinical experience levels, or specific practicum settings are not provided, which could influence safety behaviors and perceptions.

Suggestions for Improvement:

• Expand the sample size and include diverse institutions to enhance generalizability.

• Incorporate longitudinal or interventional studies to assess the impact of targeted education on safety behaviors.

• Use mixed methods combining quantitative and qualitative data for richer insights.

• Explore the influence of specific variables (e.g., prior training, workload) on safety behaviors.

Overall Assessment: The study makes a valuable contribution by identifying priority areas for nursing safety education among students. Its multi-faceted analytical approach provides practical guidance. Addressing noted limitations could strengthen future research and implementation efforts to improve nursing education and safety practices.

Reviewer #2: Questions for Authors:

1- Please clarify the measurement scales and anchors used for importance vs performance. Were the anchors identical or tailored per construct?

2- In Table 2, for “Wear a mask when having respiratory symptoms,” the reported gap (0.50 ± 0.22) appears inconsistent with the means (3.98 vs 3.93). Is this a typographical error? If so, please correct and update related statistics (t, p, Borich rank).

3- In Section 3.3, Quadrants III and IV are described in ways that conflict with standard IPA definitions and your own earlier definitions. Could you correct the quadrant descriptions and verify the item assignments accordingly?

4- How were missing data handled (if any) at the item level? Please report item-wise response rates and any imputation procedures.

5- Did you compute reliability (Cronbach’s alpha) separately for importance items and performance items, and for each domain? If so, please report them; if not, could you add these analyses?

6- Was any factor analysis (EFA or CFA) conducted to validate the four-domain structure? If not, could you provide at least an EFA to support the latent structure?

7- Given 29 paired comparisons, did you adjust for multiple testing? If not, how robust are your conclusions under a correction (e.g., Holm-Bonferroni)?

8- The domain-level “perceived performance levels” in Table 1 (e.g., 4.75 for infection prevention) differ notably from the item-level means (~3.7–3.98). How were these domain-level scores derived? Are they separate global items, and if so, how should readers interpret both?

9- Could you provide a single integrated table listing: (a) the Borich top-10 items, (b) Locus for Focus quadrant placement, (c) IPA quadrant placement, and (d) the final prioritized list with clear first/second-tier priorities?

10- For items like “Know the location of spill kits” and “Use devices to reduce musculoskeletal load,” to what extent do resource availability and institutional policies constrain performance? How might you disentangle education gaps from environmental constraints?

11- Did prior safety education (88.1% reported) associate with smaller gaps? If you tested this, please share the results; if not, could you analyze it?

12- Was the instrument administered in Korean? If so, please describe translation/back-translation and include the final instrument in supplementary materials.

Reviewer #3: 1. The manuscript is well written and clearly presented.

2. The study is interesting and highly relevant to the training of nursing students in practical settings, particularly with respect to their safety and that of the patient.

3. However, Figures 1 and 2 require revision, as their content does not fully align with the narrative. To enhance reader comprehension and ensure consistency with the text, the quadrant boundaries should include appropriate titles or labels that clearly indicate the position of each quadrant.

**Do you want your identity to be public for this peer review?** For information about this choice, including consent withdrawal, please see our For information about this choice, including consent withdrawal, please see our Privacy Policy .

Reviewer #1: **Yes:** REHEMA ABDALLAHREHEMA ABDALLAH

Reviewer #2: **Yes:** Asmaa kamal Ahmed, Associate professor of nursing administration, faculty of nursing, fayoum university, Egypt.Asmaa kamal Ahmed, Associate professor of nursing administration, faculty of nursing, fayoum university, Egypt.

Reviewer #3: No

---

## [Author Response · Author response to Decision Letter 1]

30 Jan 2026

Manuscript ID: PONE-D-25-53140R1

Type of manuscript: Article

Title: Importance and implementation of safe nursing behaviors in nursing students' clinical practice: Importance–Performance Analysis (IPA), the Borich Needs Assessment model, and the Locus for Focus model

Thank you for reviewing this study. We have attempted to address the reviewer's comments and improve the manuscript. We have included point-by-point responses to each comment in this resubmission. The detailed responses to each comment are as follows:

Response to Reviewer 1 Comments.

Weaknesses and Limitations:

1. Sample and Generalizability: The sample is limited to 160 students from two universities in South Korea, which may limit broader applicability.

Reply:

- We acknowledge that the sample was limited to nursing students from two universities in South Korea, which may affect the generalizability of the findings. This limitation has been explicitly stated in the Limitations section, and caution in interpreting and generalizing the results has been noted. We also added a recommendation for future studies to include nursing students from multiple regions and diverse institutional settings to enhance external validity.

Design and Participants

In accordance with the national nursing education accreditation standards in South Korea, all nursing students are required to complete hospital-based clinical practicums during their third and fourth academic years and to receive mandatory safety education prior to clinical practice, including training in infection prevention and patient safety.

Limitations and strengths

This study had several limitations. First, participants were limited to nursing students from two universities in South Korea. Which may restrict the generalizability of the findings. Therefore, the results should be interpreted with caution when applied to nursing students who have completed clinical practicums in a wider range of institutional and clinical settings, including diverse hospital environments and nursing education contexts. Future research should include nursing students from multiple regions and a broader range of hospitals to enhance the external validity of these findings.

2. Cross-Sectional Design: The design captures perceptions and behaviors at a single point, limiting insights into causality or changes over time

Reply:

- To address this concern, we clarified the limitations of the cross-sectional design in the Limitations section and added recommendations for future longitudinal or intervention-based studies to examine temporal changes and evaluate the effectiveness of targeted educational programs.

Limitations and strengths

Second, this study employed a cross-sectional design, which restricts causal inference and limits the examination of changes in safe nursing behaviors over time. Future studies adopting longitudinal or intervention-based designs are needed to examine temporal changes and to evaluate the effectiveness of targeted educational programs.

3. Self-Reported Data: Reliance on self-administered questionnaires may introduce response bias, with students potentially overestimating their safety behaviors.

4. Lack of Qualitative Insights: The study might benefit from qualitative data to explore underlying reasons for gaps in safety behaviors or perceived importance.

5. Limited Contextual Detail: Details such as students' prior safety training, clinical experience levels, or specific practicum settings are not provided, which could influence safety behaviors and perceptions.

Suggestions for Improvement:

• Expand the sample size and include diverse institutions to enhance generalizability.

• Incorporate longitudinal or interventional studies to assess the impact of targeted education on safety behaviors.

• Use mixed methods combining quantitative and qualitative data for richer insights.

• Explore the influence of specific variables (e.g., prior training, workload) on safety behaviors.

Reply:

- Regarding the reliance on self-reported data, the lack of qualitative insights, and limited contextual detail, we clarified in the Limitations section that the use of self-administered questionnaires may introduce response and social desirability bias and may not fully capture the complexity and contextual factors underlying nursing students’ safety behaviors. We also acknowledged that qualitative data and certain contextual variables—such as prior safety training, workload, or specific practicum settings—were not examined in depth. Accordingly, we recommended future mixed-methods studies integrating quantitative and qualitative approaches to provide a more comprehensive and context-sensitive understanding of safe nursing behaviors.

Limitations and strengths

Second, study data were collected using a self-administered questionnaire, which may be subject to response bias and social desirability bias and may not fully capture the complexity and contextual nuances of nursing students’ experiences and perceptions. In addition, several performance-related items—such as knowledge of spill kit locations and use of assistive devices to reduce musculoskeletal load—may have been influenced by institutional resource availability and organizational policies, factors that were not directly measured in this study. As a result, it was not possible to fully disentangle educational gaps from environmental or structural constraints. Future studies employing mixed-methods approaches, including qualitative interviews, direct observation, or assessment of institutional safety resources and policies, are recommended to provide a more comprehensive and context-sensitive understanding of safe nursing behaviors among nursing students.

Manuscript ID: PONE-D-25-53140R1

Type of manuscript: Article

Title: Importance and implementation of safe nursing behaviors in nursing students' clinical practice: Importance–Performance Analysis (IPA), the Borich Needs Assessment model, and the Locus for Focus model

Thank you for reviewing this study. We have attempted to address the reviewer's comments and improve the manuscript. We have included point-by-point responses to each comment in this resubmission. The detailed responses to each comment are as follows:

Response to Reviewer 2 Comments.

Weaknesses and Limitations:

1. Please clarify the measurement scales and anchors used for importance vs performance. Were the anchors identical or tailored per construct?

Reply:

- We clarify that the same safe nursing behavior instrument was used to assess both perceived importance and performance, with identical items and response anchors. The identical set of 29 items was administered twice—once for importance and once for performance—using the same five-point Likert scale with identical response anchors (1 = “strongly disagree” to 5 = “strongly agree”). This approach allowed direct comparison between the two constructs and ensured methodological consistency in subsequent analyses, including Importance–Performance Analysis, the Borich Needs Assessment model, and the Locus for Focus model. The Material and Methods section has been revised to explicitly describe this measurement approach.

Measures

Safe nursing behaviors

The same set of 29 items was used to assess both the perceived importance and the performance of safe nursing behaviors. For each item, importance and performance were measured separately using identical five-point Likert scales with the same response anchors (1 = “strongly disagree” to 5 = “strongly agree”), allowing direct comparison between the two constructs. Higher scores indicated greater perceived importance or higher levels of performance in safe nursing behaviors. Reliability analyses were conducted separately for importance and performance items for the total scale and each domain, and acceptable to high internal consistency was observed (Supplementary Table S3).

2. In Table 2, for “Wear a mask when having respiratory symptoms,” the reported gap (0.50 ± 0.22) appears inconsistent with the means (3.98 vs 3.93). Is this a typographical error? If so, please correct and update related statistics (t, p, Borich rank).

Reply:

- Thank you for identifying this inconsistency. Upon re-examination, we confirmed that the reported mean scores for this item (Importance = 3.98, Performance = 3.93) were correct, and that the originally reported gap value (0.50 ± 0.22) resulted from a decimal-point error during manuscript preparation. The corrected importance–performance gap is 0.05 ± 0.22, and the corresponding t value, p value, and Borich Needs Assessment ranking have been recalculated and updated accordingly. These corrections did not change the overall pattern of results or the study’s main conclusions. All related values in Table 2 and the corresponding text have been revised. We appreciate the reviewer’s careful reading.

3. In Section 3.3, Quadrants III and IV are described in ways that conflict with standard IPA definitions and your own earlier definitions. Could you correct the quadrant descriptions and verify the item assignments accordingly?

Reply:

- Thank you for pointing out this inconsistency. We carefully reviewed Section 3.3 and revised the descriptions of Quadrants III and IV to ensure full consistency with standard Importance–Performance Analysis (IPA) definitions and our earlier methodological descriptions. Quadrant III is now defined as items with low importance and low performance (Low Priority), whereas Quadrant IV is defined as items with low importance but relatively high performance (Possible Overkill). Following these revisions, all item assignments were rechecked and confirmed to be consistent with the corrected quadrant definitions. The revised descriptions and verified item placements have been incorporated into the updated manuscript.

Importance–Performance Analysis

• Quadrant 3 (LP) contains items rated low in terms of both importance and performance. These comprised Items 1, 6, 14, 16, and 18–20. This indicates that these behaviors were perceived as less important and were also infrequently performed, suggesting a relatively low priority for immediate intervention.

• Quadrant 4 (PO) includes items with lower importance but relatively higher performance. These were: “Handwashing before leaving the patient room” (Item 4), “Avoid contamination when handling soiled linen” (Item 8), and “Positive engagement with peers” (Item 22). This pattern suggests that these behaviors may be performed more than warranted based on their perceived importance, indicating a potential over-allocation of effort or resources.

This IPA matrix provides clear guidance for prioritizing efforts to enhance safe nursing behaviors among nursing students, emphasizing which performance strengths should be maintained and which critical gaps should be addressed (Fig. 1).

4. How were missing data handled (if any) at the item level? Please report item-wise response rates and any imputation procedures.

Reply:

- Data were collected using an online survey system that required responses to all items before submission; therefore, no item-level missing data occurred. The item-wise response rate was 100%, and no imputation procedures were applied. This has been clarified in the Material and Methods section of the revised manuscript.

Design and Participants

Data collection was conducted using an online survey platform configured to require completion of all items prior to submission; consequently, no item-level missing data were observed, and all collected responses were included in the final analysis.

5. Did you compute reliability (Cronbach’s alpha) separately for importance items and performance items, and for each domain? If so, please report them; if not, could you add these analyses?

Reply:

- Yes, reliability analyses were conducted separately for importance and performance items, both for the overall scale and for each domain. The results are summarized in Supplementary Table S3. Internal consistency was high for the total scale (importance: Cronbach’s α = 0.90; performance: Cronbach’s α = 0.92). At the domain level, Cronbach’s alpha coefficients ranged from 0.55 to 0.83 for importance and from 0.69 to 0.84 for performance.

The musculoskeletal injury prevention domain, which consists of three items, showed a relatively lower alpha for importance (α = 0.55), likely reflecting the small number of items and the known sensitivity of Cronbach’s alpha to item count. Domains with a larger number of items demonstrated acceptable to good internal consistency. Given that the primary aim of this study was needs assessment and prioritization rather than psychometric scale development, these reliability levels were considered acceptable for exploratory and applied research. The corresponding results have been added to the revised manuscript.

Safe nursing behaviors

Reliability analyses were conducted separately for importance and performance items for the total scale and each domain, and acceptable to high internal consistency was observed (Supplementary Table S3).

6. Was any factor analysis (EFA or CFA) conducted to validate the four-domain structure? If not, could you provide at least an EFA to support the latent structure?

Reply:

- Exploratory or confirmatory factor analysis (EFA/CFA) was not conducted in the present study. The four-domain structure of the instrument was defined a priori based on an extensive literature review and the Korean Occupational Safety and Health Guide, and content validity was established through expert panel review rather than empirical factor extraction.

The primary aim of this study was needs assessment and priority setting using Importance–Performance Analysis (IPA), the Borich Needs Assessment model, and the Locus for Focus model, rather than psychometric scale development or validation. Accordingly, the instrument was treated as a multidimensional checklist of conceptually distinct safety-related behaviors. In addition, some domains consisted of a small number of items, which limits the interpretability and stability of factor analytic results. For these reasons, an exploratory factor analysis was not additionally conducted in the current study.

Nevertheless, we agree that further psychometric evaluation would strengthen the instrument. Future studies with larger and more diverse samples should consider conducting EFA or CFA to further examine the latent structure and dimensionality of the scale. This point has been added to the Limitations section of the revised manuscript.

Limitations and strengths

Fifth, exploratory or confirmatory factor analyses (EFA/CFA) were not conducted to empirically validate the four-domain structure, as the instrument was theoretically defined a priori and content validity was established through expert review. Future studies with larger and more diverse samples are warranted to examine the latent structure of the instrument using factor analytic approaches.

7. Given 29 paired comparisons, did you adjust for multiple testing? If not, how robust are your conclusions under a correction (e.g., Holm-Bonferroni)?

Reply:

- No formal adjustment for multiple testing was applied in the primary analysis, as the item-level paired t-tests were used for descriptive and exploratory purposes to inform the priority-setting frameworks (IPA, Borich Needs Assessment, and the Locus for Focus model), rather than for confirmatory hypothesis testing.

To address the reviewer’s concern, we conducted a sensitivity analysis using the Holm–Bonferroni correction. After adjustment, the overall pattern of importance–performance differences remained unchanged, and the items identified as high-priority across the three analytic frameworks were the same. The uncorrected and adjusted p-values are presented in Supplementary Table S2. These results indicate that the study’s conclusions are robust to multiple testing correction.

Statistical Analysis

In addition, a sensitivity analysis using the Holm–Bonferroni correction was conducted to assess the robustness of the item-level paired comparisons, and the results are presented in Supplementary Table S2.

8. The domain-level “perceived performance levels” in Table 1 (e.g., 4.75 for infection prevention) differ notably from the item-level means (~3.7–3.98). How wer

---

## [Editor Report · Decision Letter 1]

25 Feb 2026

Importance and implementation of safe nursing behaviors in nursing students' clinical practice: Importance–Performance Analysis (IPA), the Borich Needs Assessment model, and the Locus for Focus model

PONE-D-25-53140R1

Dear Dr. Hae Ran Kim,

We’re pleased to inform you that your manuscript has been judged scientifically suitable for publication and will be formally accepted for publication once it meets all outstanding technical requirements.

Kind regards,

Priti Chaudhary, M.S.

Academic Editor

PLOS One
---

## [Editor Report · Acceptance letter]

PONE-D-25-53140R1

PLOS One

Dear Dr. Kim,

I'm pleased to inform you that your manuscript has been deemed suitable for publication in PLOS One. Congratulations! Your manuscript is now being handed over to our production team.

Kind regards,

on behalf of

Dr. Priti Chaudhary

Academic Editor

PLOS One